

# Evaluation of hygroscopic cloud seeding in warm rain process by a hybrid microphysics scheme on WRF model: a real case study

Kai-I Lin[1], Kao-Shen Chung[1,*], Sheng-Hsiang Wang[1], Li-Hsin Chen[2], Yu-Chieng Liou[1], Pay-Liam Lin[1], Wei-Yu Chang[1], Hsien-Jung Chiu[1], Yi-Hui Chang[2]

[1]Department of Atmospheric Sciences, National Central University, Taoyuan, Taiwan
[2]National Chung-Shan Institute of Science and Technology, Taiwan

*Correspondence to*: kschung@atm.ncu.edu.tw, kaoshen.chung@gmail.com

**Abstract.** To evaluate the hygroscopic cloud seeding in reality, this study develops a hybrid microphysics scheme on WRF model, WDM6–NCU, which involves 43 bins of seeded cloud condensation nuclei (CCN) in the WDM6 bulk method scheme.

This scheme can describe the size distribution of seeded CCNs and explain the process of the CCN imbedding, cloud and raindrop formation in detail. Furthermore, based on the observational CCN size distribution applied in the modelling, a series of tests on cloud seeding was conducted during the seeding periods of 21–22 October, 2020 with stratocumulus clouds. The model simulation results reveal that seeding at in-cloud regions with an appropriate CCN size distribution can yield greater rainfall and that spreading the seeding agents over an area of 40–60 km$^2$ is the most efficient strategy to create a sufficient

precipitation rate. With regard to the microphysical processes, the main process that causes the enhancement of precipitation is the strengthening of the accretion process of raindrops. In addition, hygroscopic particles larger than 0.4 μm primarily contribute to cloud-seeding effects. The study results could be used as references for model development and warm cloud seeding operations.

## 1 Introduction

Global warming has made droughts much more frequent (Bo-Tao, 2021). In 2021, for the first time in a century, Taiwan experienced the most severe drought, which prompted the Taiwanese government to identify methods to address water scarcity problems with utmost urgency. Cloud seeding, a common method of weather modification, appears to be a possible means to creating more water resources. According to Lelieveld (1993), 80% of cloud droplets are unable to reach the ground, which indicates inefficiency in the transformation of cloud droplets to raindrops. Thus, since the 20th century, cloud seeding research

has been expanding. However, no single cloud-seeding strategy can be applied worldwide because environmental conditions differ between countries. Two main strategies of cloud seeding are frequently adopted: ice cloud seeding (Seto et al., 2011;Geresdi et al., 2017;Tessendorf et al., 2019;Wang et al., 2021) and warm cloud seeding (Jung et al., 2015;Wang et al., 2019;Tonttila et al., 2021;Tessendorf et al., 2021).

During the dry season (i.e., October to April) in Taiwan, clouds tend to be warm and relatively thin, with a cloud base of

approximately 500 m above mean sea level (Chen, 1995). In addition, due to the prevailing northeasterly wind, weather systems



persist for a long time and large amounts of water vapour are supplied. Therefore, warm cloud seeding is more appropriate for use in Taiwan. Hygroscopic cloud seeding is a type of warm cloud seeding and has been used in Taiwan. In warm clouds, giant cloud condensation nuclei (GCCN: diameter > 1 μm) in turn increase the mean droplet diameter and increase the precipitation amount (Lehahn et al., 2011;Dadashazar et al., 2017;Feingold et al., 1999;Jensen and Lee, 2008;Jensen and

Nugent, 2017). The seeding agents used in hygroscopic cloud seeding serve as efficient CCNs or GCCNs and play a crucial role in strengthening the condensation and collision–coalescence process, thereby widening the droplet size distribution (DSD) and increasing the precipitation efficiency (Jensen and Lee, 2008;Jung et al., 2015;Tessendorf et al., 2021). The process from the spreading of seeding agents to development of rainfall takes approximately 10–20 min (Silverman, 2000;Tonttila et al., 2021). However, despite the availability of clearly elaborated theories related to hygroscopic cloud seeding, scientific evidence

on the seeding effects and the efficiency of the strategy are scarce.

The effects of cloud seeding have been mainly verified using a statistical approach, which is based on the comparison of multiple observational samples with seeding and nonseeding scenarios (Gagin and Neumann, 1981;Silverman, 2000). However, this method may involve high uncertainty due to the difficulties of conducting a long-term, consistent, and randomized cloud-seeding experiments (Guo et al., 2015;Wang et al., 2019). Recently, due to advances in observation methods,

a greater variety of instruments, such as cloud radars and cloud droplet probes, have been used to investigate cloud-seeding effects and obtain direct observational evidence. Until now, comprehensive evidence has not fully investigated because determining whether the precipitation signal is due to cloud seeding or to meteorological variations is difficult (Kerr, 1982;Mather et al., 1997;Silverman, 2003;Flossmann et al., 2019;Tonttila et al., 2021).

Model simulation can be used to efficiently generate several realizations of each scenario and has the advantage of

separating the cloud-seeding signal from its natural counterpart. Caro et al. (2002) concluded that hygroscopic particles with a radius between 0.5 and 6 μm are optimal for enhancing precipitation in warm clouds. Segal et al. (2004) reported that hygroscopic cloud seeding with particles with diameters of 1.5–2.5 μm lead to considerable increment of precipitation. Cloud seeding with giant CCNs strengthens the autoconversion and the accretion process, leading to an enhancement of precipitation (Tonttila et al., 2021). However, most studies on warm-cloud-seeding simulation have performed simulations by using a one-

dimensional cloud parcel model or by using ideal cases (Cooper, 1997;Caro et al., 2002;Segal et al., 2004;Chen et al., 2020;Tessendorf et al., 2021;Tonttila et al., 2021), which may not accurately reflect actual environmental conditions.

The present study evaluated the effects of cloud seeding on a realistic environment of northern Taiwan by using a Weather Research and Forecasting (WRF) model with a hybrid microphysics scheme to yield more accurate results of cloud seeding without incurring overly large computational costs. In addition, the effective strategies were mapped by conducting a series of

cloud-seeding sensitivity tests. The remainder of the article is organized as follows. Section 2 describes the main characteristics of microphysics schemes, WDM6 and WDM6–NCU, and Section 3 presents the modelling settings and experiment design. Finally, Sections 4 and 5 present the study results and a discussion.



## 2 Model description

This study uses the fully compressible and nonhydrostatic WRF model version 3.9.1 to simulate three-dimensional
meteorological parameters. The WRF model employs an eta coordinate, which allows the grids to follow the complex terrain,
and the third-order Runge–Kutta numerical method for solving the time split integration of the governing equation. In addition,
the Arakawa C-grid is used in the simulations, which leads to the arrangement of thermal parameters at the centre grids and
that of wind speed variables at the staggered grids. With regard to CCNs and cloud microphysics, the WDM6–NCU
microphysics scheme, which is modified from WDM6 by NCU, is used to represent the properties of CCNs, cloud, and rain.

**2.1 WDM6 (Lim and Hong, 2010)**

WDM6 is a semi-double-moment bulk microphysics scheme, which predicts not only the mixing ratio of the hydrometeors
such as cloud droplets and raindrops but also their number concentrations. The cloud–raindrop size distribution is given as
follows:

$$n_x = N_x \frac{\alpha_x}{\Gamma(\mu_x)} \lambda_x^{\alpha_x \mu_x} D_x^{\alpha_x \mu_x - 1} e^{-(\lambda_x D_x)^{\alpha_x}}, \tag{1}$$

where $x$ represents the type of hydrometeor, including clouds and rain. $\lambda_x$, $\mu_x$, and $\alpha_x$ are slope parameter and two dispersion
parameters, respectively, and $N_x$ and $D_x$ represent the predicted value of the total number concentration and diameter of the
certain hydrometeor category. Moreover, the dispersion parameters of rain $\mu_R$ and $\alpha_R$ are set as 2 and 1, which provides the
advantage of simulating a more reasonable raindrop shape and size distribution.

Moreover, to evaluate the effect of CCNs, the relationship between the number of activated CCN ($n_a$) and the
supersaturation ($S_w$) is used in WDM6 as follows (Twomey's relationship):

$$n_a = (n + N_c)(\frac{S_w}{S_{max}})^k, \tag{2}$$

where $n$, $Nc$, and $S_{max}$ are the total CCN number concentration, cloud droplet number concentration, and supersaturation
required to activate the total particle count, respectively. In Eq. (2), $k$ typically ranges from 0.3 to 1.0. In addition, the
production rate for the cloud water mixing ratio by CCN activation ($P_{cact}$) can be expressed as

$$P_{cact} = \frac{4\pi \rho_w}{3 \rho_a} r_{act}^3 \times n_a, \tag{3}$$

where $\rho_w$ and $\rho_a$ are the density of water and air, respectively. In Eq. (3), $r_{act}$ is the radius of the activated droplets, which is set
at a fixed value (1.5 µm) in WDM6. Microphysics schemes are seldom able to describe CCN effects, and this advantage of the
WDM6 scheme enables the simulation of cloud seeding in a more realistic environment with a specific CNN size.



## 2.2 WDM6–NCU

The effects of cloud seeding are highly dependent on the CCN size distribution (Caro et al., 2002;Segal et al., 2004). Therefore, in the improved WDM6–NCU microphysics scheme with the bin-resolving method, the seeded CCNs are described using bins of 43 sizes to evaluate the effects of CCN. Figure 1 displays the schematic of the two methods used to describe aerosol size distribution. In addition, the size distribution of the seeded CCNs is based on observation and is fitted into a trimodal lognormal function as follows:

$$\frac{dN}{d\,lnr_n} = \sum_{i=1}^{3} \frac{n_i}{\sqrt{2\pi}\,log\sigma_i\,ln10} \exp\left[-(\frac{lnr_n - lnR_i}{\sqrt{2}\,ln\sigma_i})^2\right],$$   (4)

where $r_n$, $n_i$, $R_i$, and $\sigma_i$ are the radius of the particle, total number concentration, geometric mean radius, and geometric standard deviation for each mode (indicated by subscript $i$), respectively. The complete CCN size distribution can be used to accurately calculate the critical radius based on Köhler theory (Köhler, 1921), and the bins of CCNs whose size extends the critical radius will be able to activate the corresponding liquid bins. The critical radius ($r_{cr}$) is expressed as follows:

$$r_{cr} = \frac{A}{3}\left(\frac{4}{BS_w^2}\right)^{1/3},$$   (5)

where $A$ is the parameter related to temperature, $B$ is the parameter whose value differs between chemicals, and $S_w$ represents the supersaturation ratio. After the number concentration and mixing ratio of the liquid bins are calculated, they are used in the calculation of the mixing ratio and number concentration of cloud and rain, and the microphysics processes continue as is the case in the original WDM6. Figure 2 illustrates the cloud-seeding effects on droplet size as model applying the bin and bulk methods for cloud seeding with the same number concentration (200 # cm⁻³). The GCCN size distribution of Chemical

Systems Research Division (CSRD: shown in section 3.2) was seeded by bin method, and fixed size CCNs with 1.2 μm (the mean diameter of GCCNs in CSRD) were seeded by bulk method. Results show that the mean-volume-drop diameter of cloud (Dc) and mean-volume-drop diameter of rain (Dr) are considerably larger than those obtained in seeding as modelled using the bulk method. This is because in contrast to the bulk method, the bin method accounts for the complete seeded CCN size

distribution but does not have a fixed CCN size. Thus, the WDM6–NCU accounts for CCN effects to yield more accurate results without incurring overly large computational costs.

## 3 Model setup

We implement hygroscopic cloud seeding with the stratocumulus clouds resulting from the stronger northeasterly winds as our target. In this study, the hygroscopic cloud-seeding effects are investigated for the case of 21–22 October, 2020. On 21–

22 October, 2020, the typhoon Saudel, was located in southwestern Taiwan, and accompanied the comovement of the northeast monsoon, which caused the occurrence of stronger northeasterly winds and brought large amounts of water vapour to northern Taiwan. According to the weather map and a skew-T diagram, the environment was saturated below the mean sea level height of approximately 2000 m at 00:00 UTC on 22 October, 2020.



### 3.1 Model configuration

Five nested domains are constructed (Figure 3a) with 52 vertical levels below 10 hPa and horizontal resolutions of 27, 9, 3, 1, and 0.333 km corresponding to $190 \times 151$, $301 \times 250$, $301 \times 301$, $271 \times 406$, and $202 \times 202$, respectively. Moreover, the initial and boundary conditions are generated from the National Centers for Environmental Prediction Final (NCEP FNL) operational model global tropospheric analysis at a resolution of 0.25°. For the first to fourth domains (D01–D04), the simulation is integrated from 21 October, 2020, 12:00 UTC, to 22 October, 2020, 12:00 UTC, with a time step of 90, 30, 10,
and 10/3 s, respectively. However, for the fifth domain (D05), the simulation is conducted from 06:00 UTC on 22 October, 2020, to 09:00 UTC on 22 October, 2020, with a time step of 1 s.

The physical parameterizations used in this study include the rapid radiative transfer model (RRTM) longwave scheme (Mlawer et al., 1997), Dudhia shortwave scheme (Dudhia, 1989), Yonsei University (YSU) planet boundary scheme (Hong et al., 2006), Grell Devenyi ensemble cumulus scheme (Grell and Dévényi, 2002), Monin–Obukhov land surface scheme (Monin
and Obukhov, 1954), and WRF double-moment six-category scheme (Lim and Hong, 2010) modified with WDM6–NCU microphysics scheme. The cumulus scheme is only used in D01 and D02. Regarding the planet boundary scheme, YSU is used in D01–D04 and large eddy simulation (LES) is used in D05. Table 1 presents a summary of the model configurations. According to Weigel et al. (2007) and Xue et al. (2014), simulations with a high-resolution LES (resolution < 800 m) can efficiently reproduce flow characteristics over a complex terrain. Thus, the simulation conducted in this study should more
accurately reflects actual conditions.

### 3.2 Characterization of CCN size distribution

The size distribution of CCN considerably affects the cloud microphysics processes. Several studies have reported that cloud features and precipitation level are dependent on CNN size (Yin, 2000;Bruintjes, 2003;Segal et al., 2004;Rosenfeld and M., 2008;Rosenfeld et al., 2014;Guo et al., 2016;Lee et al., 2016). According to previous research, larger CCNs (>2 μm) are
optimal for increasing precipitation, while a high concentration of small CCNs can suppress rainfall or postpone the onset of precipitation. Therefore, it is crucial to treat CNN size distribution more realistically in the cloud-seeding experiment for both observation and model simulation. In this study, we conducted a chamber experiment to characterize CCN size distribution. The flare agent we used in chamber was provided by the Chemical Systems Research Division (CSRD), National Chung-Shan Institute of Science and Technology, who has been invested the flare agent for cloud-seeding operations in Taiwan.

Based on an evaluation of 200 samples of CSRD flare seeding agents by using an aerosol spectrometer (Grimm 11-D), the size distribution can be evaluated and fitted as the sum of the three lognormal modes in the model simulation, as shown in Figure 4 with the constraining parameters of each mode for the total number concentration (N), geometric mean diameter (D), and geometric standard deviation (σ) listed in the text box. In addition, the composition of the CSRD seeding agent is mainly sea salt, which is also characterized in WDM6–NCU. Thus, the simulation conducted in this study has the potential to
accurately reflect actual CCN microphysics. In our practical operation, hygroscopic cloud seeding is conducted using drones





that carry 10 CSRD flare seeding agents in 10 min, affording greater flexibility in seeding height. However, information for determining the most effective cloud-seeding operation is lacking. Therefore, in the following section, we use modelling approach, conducting a series of simulations with several scenarios of 10-min cloud-seeding processes by using 10 CSRD flare seeding agents to evaluate the cloud-seeding effects at different seeding heights, seeding areas, and seeding concentrations.

**3.3 Experimental design**

Two parallel sets of experiments, namely a control simulation without seeding (control run) and a set of experiments with the emission of seeding agents (seed run), with seeding started on 22 October, 2020, 06:30 UTC in Shihmen reservoir (latitude, 24.81 ° N; longitude, 121.26 ° E) and with the CSRD size distribution, were designed to analyse the effects of aerosol perturbation. In the seed run, we further examine the effects of cloud seeding at different heights and in different areas of the Shihmen region (Figure 3b). In domain four (D04; 1-km horizontal resolution), five simulations are formulated, namely one control run (Ctrl) and four seed runs (Seed 1 to Seed 4), to investigate the effects of cloud seeding in one horizontal grid (1 km × 1 km) but at different seeding levels. In Seed 1, the cloud seeding is conducted at approximately 500 m above mean sea level, which is close to the cloud base according with sounding and model simulation data. Seeds 2 and 3 involve simulated seeding between 1000 and 2000 m, and Seed 4 involves seeding at approximately 2200 m. The model experiments in D04 are summarized in Table 2.

Several studies have investigated cloud-seeding effects through simulations at a finer grid resolution (Yin, 2000;Tonttila et al., 2021;Xue et al., 2014). To further examine and interpret the effects of cloud seeding and microphysical processes, domain five (D05) is developed at a horizontal resolution of 333 m. The seeding heights in D05 are based on the results of D04, and two seeding heights that yield the smallest and greatest increases in rainfall in D04 (500 m and 1300 m) are selected. Six runs are developed to seed hygroscopic particles in different areas, namely 1, 10, and 100 km$^2$, at the selected seeding levels. In addition, two more runs with 100 times the concentration of seeding agents in a 1-km$^2$ area are performed for the seeding height at 500 m and 1300 m, respectively. The model experiments in D05 are summarized in Table 3.

**4 Results**

**4.1 Control run in D04 (Ctrl_D04)**

Before the cloud-seeding simulation assessment, results from a control run are validated against the observations to ensure that the simulation accords reasonably closely with reality, and we take accumulated rainfall, radar reflectivity, water vapour mixing ratio, temperature, and pressure into consideration. The main features of cumulative rainfall (Figure 5) are successfully captured by the model, particularly in northern Taiwan, our location of interest. Moreover, information on observational radar reflectivity is used to account for precipitation patterns at different heights (Figure 6). The radar reflectivity results show that most of the reflectivity is below 5 km. This indicates that the rainfall system mainly involves warm rain processes, and this feature is also captured by the model simulation. Furthermore, the temperature, pressure, and water vapour mixing ratio are



similar between the observational data in the Dongyan mountain site, located near Shihmen region, and the corresponding grid-point data in the model simulation (Figure 7). Thus, the simulation conducted in this study can be used for tests of cloud-seeding sensitivity of the WRF model with the new hybrid scheme. Figure 8 presents the meridional mean (0.1° latitude across

Shihmen) of liquid water content (LWC) at 06:30 UTC on 22 October, 2020. In Figure 8, it shows that cloud is approximately located above 500 m and below 2000 m in the Shihmen region with the highest LWC appears near 1300 m, and all of the clouds are below 5 ˚C line, which represents the warm rain processes are dominated.

**4.2 Seeding effects on precipitation and cloud properties**

To determine the effects of cloud seeding at different seeding heights, four seed runs (Seed1 to Seed4) in one horizontal grid

(1 km × 1 km) at different seeding levels (~500, 1000, 1300, and 2200 m above mean sea level) are executed in D04. An evaluation of the average rain rate of seed runs and the control run in the Shihmen region (Figure 9) indicates that seeding above 1000 m but below 2000 m enhances surface rainfall in the Shihmen region, and Seed3 resulted in the greatest enhancement of precipitation. The results also indicate that seeding at the in-cloud levels (Seed 2 and Seed 3) is more effective for precipitation than seeding at cloud base (Seed1) in northern Taiwan. Thus, in this study, seeding at 500 and 1300 m,

corresponding to Seed1 and Seed3, are chosen as the runs that yielded the lower and higher rainfall enhancement for the further simulation assessments and data analysis in domain five (D05).

In D05, eight runs are executed at different seeding heights (500 and 1300 m), but for different seeding areas (1, 10, and 100 km$^2$), and seeding concentrations. As shown in Figure 10, seeding at 1300 m yields greater rainfall than seeding at 500 m, and this result is similar to that for domain four (D04). In addition, seeding at 1300 m, over a bigger seeding area, and at higher

aerosol concentrations yielded greater rain rates by several folds in the Shihmen region (Figure 10), particularly in seeding areas of 100 km$^2$. However, this phenomenon is not obvious in scenarios of seeding at 500 m, suggesting in-cloud seeding is more beneficial compared to below cloud seeding. If cloud seeding is conducted in the appropriate environments that afford, for example, bigger seeding areas or in areas with higher aerosol concentrations, precipitation is enhanced.

In terms of microphysical properties of simulation, a peak in seeded CCN concentration is observed at their seeding height,

500 and 1300 m (Figure 11a–c). In the scenario of seeding at 1300 m (Figure 11a–c: warm colour), the distribution of the seeded CCNs is able to transport to higher altitude than the scenarios seeding at 500 m (Figure 11a–c: cold colour). Figure 11d–f shows that both seeding at 500 and 1300 m enhance the mixing ratio of cloud (QCLOUD) in 10 min (from 06:30 to 06:40) after cloud seeding. However, QCLOUD starts to decrease after 06:40 (10 min after cloud seeding) in the seeding scenario at 1300 m, but this phenomenon is not apparent in the seeding scenario at 500 m. With regard to the mixing ratio of

rain (QRAIN), Figure 11g–i indicates that seeding at 1300 m considerably increases QRAIN after cloud seeding, but seeding at 500 m only weakly enhances QRAIN. This phenomenon also explains why QCLOUD starts to decrease after 10 min of seeding at 1300 m, but this behaviour is not obvious in the seeding scenario at 500 m. In addition, by using the double-moment microphysics scheme, we calculate the mean-volume-drop diameter of rain ($Dr$) as




$$\lambda_r = (\frac{4\pi\rho_w N_r}{\rho_a q_r})^{\frac{1}{3}} ; D_r = \frac{1}{\lambda_r}(24)^{1/3}, \tag{6}$$

where $N_r$, $Q_r$, $\rho_w$ and $\rho_a$ are the number concentration of rain, mixing ratio of rain, density of water, and density of air, respectively. Figure 11j–l shows that $D_r$ increases more obviously in the scenarios of seeding at the in-cloud region (the warm colour lines). If larger raindrops develop, more liquid water may reach the ground, increasing surface rainfall. Finally, the model reveals a slight change in supersaturation ratio between the experiments (Figure 12), indicating that although the seeding agents can compete for water vapour, only a little amount of water vapour is consumed, and therefore, the saturation state of

the environment is not highly affected (approximately fluctuate 0.5 %) by cloud seeding. According to Tonttila et al. (2021), the ideal simulation of hygroscopic cloud seeding presents that water vapour competition does not cause the great seeding effects on cloud supersaturation. In this study, when the supersaturation ration is relative high in the environment, similar results can be found by the complex simulation.

**4.3 Seeding effects on microphysical processes**

Five microphysical parameters are considered, namely cloud activation ($P_{cact}$), cloud condensation ($P_{cond}$), evaporation of rain ($P_{revp}$), autoconversion of rain ($P_{raut}$), and accretion of rain ($P_{racw}$). We integrate the simulations at heights below 5 km to obtain the averaged difference between the control run and seed runs for each parameter. As shown in Figure 13, 10 min after cloud seeding, the seeding effect is mainly observed in terms of $P_{cact}$, $P_{raut}$, and $P_{racw}$. For $P_{cact}$, as depicted in Figure 14a–c, the activation process is intense at the height at which the seeding agents are introduced; this mainly occurs in 10 min after cloud

seeding. Moreover, because the supersaturation ratio at 1300 m is higher than that at 500 m, seeding at 1300 m yields a stronger $P_{cact}$ than seeding at 500 m. Regarding $P_{raut}$, the seeding scenarios in areas of 100 km² at 500 m and 1300 m exhibit obvious but opposite signals. As shown in Figure 14d–f, the autoconversion process is clearly stronger in Seed_500 (100 km²) but weaker in Seed_1300 (100 km²) 15 min after cloud seeding. However, with regard to $P_{racw}$, the seeding scenarios at 1300 m (Figure 14g–i: warm colour) yield a more intense accretion process than the scenarios at 500 m (Figure 14g–i: cold colour).

Thus, in our model simulation, introducing seeding agents with a CSRD size distribution can enhance the activation process ($P_{cact}$), and seeding at 1300 m can promote the activation of more seeded CCNs into clouds. In addition, because of the strengthening of accretion process ($P_{racw}$), more precipitation can be developed in the seeding scenarios at 1300 m. Tonttila et al. (2021) also shows the enhanced accretion process is the main pathway for precipitation enhancement after cloud seeding. At the seeding scenarios at 500 m, rainfall is slightly enhanced, mainly due to the enhancement of the autoconversion process

($P_{raut}$) when a seeding agent is introduced in an area of 100 km²; however, seeding in such a large domain can be impractical and ineffective. Therefore, cloud seeding at 1300 m (in-cloud area) seems to be the more suitable choice for increasing rainfall. Figure 15 presents more details of the effect of cloud seeding on the cloud microphysical properties by seeded CCN size distribution and time in Seed_1300(100 km²). Figure 15 also indicates that the fraction of hygroscopic particles larger than 0.4 μm decreases over time. This phenomenon indicates that particles larger than 0.4 μm are the main factor contributing to cloud-

seeding effects.



## 5. Discussion and suggestions

For hygroscopic cloud seeding practice, two crucial questions are often be asked: 1) which types of environmental conditions are appropriate for executing hygroscopic cloud seeding in the stratiform system? and 2) what the optimal seeding area is? Based on the results of the study, seeding at in-cloud levels can enhance precipitation more than seeding at the cloud base because of an enhanced accretion process. Distributing the hygroscopic particles into larger areas in clouds is more effective in enhancing rainfall. More detailed discussions are followed.

With regard to the first question, an altitude of 1300 m above mean sea level (the in-cloud region), which can increase rainfall the most in the simulation after cloud seeding, is used as a reference. In this case, the cloud base is at approximately 500 m, and the supersaturation ratio is in the range of 1.5%–2% near the altitude of 1300 m in the Shihmen region. However, after cloud seeding, the supersaturation ratio can be approximately consumed 0.5% through water vapour competition, while the LWC is approximately 0.6 $g^{-1}$ g $m^{-3}$. Therefore, our recommendation for the stratiform system, hygroscopic particles should be introduced into the in-cloud region where the supersaturation ratio is more than 0.5% and LWC is higher than 0.6 g $m^{-3}$. With regard to the second question, to determine a practicable seeding area that yields increased precipitation, two more cloud seeding runs, over seeding areas of 36 and 64 $km^2$ at an altitude of 1300 m, are developed. Figure 16 displays the average rain enhancement rate in the Shihmen region in 20 min in scenarios with different seeding areas, and the results show that when the seeding area is smaller than 64 $km^2$, the rain rate is obviously enhanced. However, for seeding areas larger than 64 $km^2$, a slight increase in rain rate is observed because the Shihmen region no longer has plenty of cloud water to transform to precipitation (Figure 16). Thus, we recommend spreading the seeding agents over an area of 40–60 $km^2$ because it can be feasibly used to yield the greatest rainfall.

## 6 Conclusion

In this study, the WRF model with the WDM6–NCU microphysics scheme, which can describe the seeded CCN size distribution with 43 bins and precisely evaluate the activation of seeded CCNs, is developed and used to simulate the case of 21–22 October, 2020. A realistic dataset describing size distribution of the flare agent is used in the model simulation. In D04, one control run and four seed runs (Seed 1 to Seed 4) are conducted in one horizontal grid (1 km × 1 km) at different seeding levels (~500, 1000, 1300, and 2200 m above the mean sea level). The results reveal that seeding above 1000 m but below 2000 m enhances cumulative rainfall in the Shihmen region for 1 hour after cloud seeding, and seeding at 500 m (cloud base) and 1300 m (in-cloud region), corresponding to Seed 1 and Seed 3, are selected as the runs with the lower and upper bound in rainfall, respectively, for subsequent sensitivity assessments and analyses (D05: 333 m × 333 m in horizontal resolution).

In D05, eight runs are developed to examine the effect of cloud seeding at different seeding heights (500 and 1300 m), seeding areas (1, 10, and 100 $km^2$), and different seeding concentrations. With regard to the sensitivity of precipitation, the model simulation reveals that more precipitation is observed at the seeding scenarios at the in-cloud region and that introducing hygroscopic particles into a bigger domain or with higher concentrations can increase precipitation by several folds in the in-





cloud-seeding simulations. Moreover, the seeding scenarios at different heights have different microphysical properties. First, seeding at 1300 m can transport seeded CCNs to higher levels and lead to a thicker CCN vertical distribution than the seeding
scenarios at 500 m. Second, both seeding at 500 and 1300 m can enhance mixing ratio of cloud (QCLOUD) within 10 min after cloud seeding; however, QCLOUD decreases earlier in the seeding scenarios at 1300 m because more cloud droplets can turn into raindrops. Third, seeding at 1300 m produces a stronger increase in mixing ratio of rain (QRAIN) than seeding at 500 m within 30 min after cloud seeding. The mean-volume diameter of raindrop (Dr) increases more obviously in the seeding scenario at 1300 m, which results in more liquid water reaching the ground, thereby enhancing surface rainfall. Moreover, the
signals are always more intense in the runs with bigger seeding domains or higher seeded CCN concentrations. Furthermore, only a few water vapour face competition from hygroscopic particles and, therefore, the saturation state of the environment is not extremely affected by cloud seeding. The seeding effects on microphysical process, primarily cloud activation ($P_{cact}$), autoconversion of rain ($P_{raut}$), and accretion of rain ($P_{racw}$) are evaluated. The results reveal that CSRD seeding agents can enhance the activation process ($P_{cact}$), and seeding at 1300 m can activate the seeding of more CCN into clouds. In addition,
because of the strengthening of the accretion process ($P_{racw}$), more precipitation is developed in seeding scenarios at 1300 m (in-cloud region). Although the seeding scenarios at 500 m and with an area of 100 $km^2$ enhanced the rainfall, mainly due to the enhancement of the autoconversion process ($P_{raut}$), the enhancement is not efficient. Finally, the size distribution of the CCNs after cloud seeding illustrates that hygroscopic particles larger than 0.4 μm primarily play an important role on cloud-seeding effects

Overall, this study develops a hybrid cloud-seeding microphysics scheme and selects a case with optimal model performance and a typical weather condition in northern Taiwan to conduct a series of cloud-seeding sensitivity tests. In addition, the study elucidates the microphysics processes that are involved from the launching of cloud seeding to the development of rainfall in northern Taiwan. In the future, more cases can be applied and statistical analysis can be conducted. Furthermore, observational verification of cloud-seeding effects should also be conducted.




*Data availability.* The radar data were provided by Central Weather Bureau, Taiwan. The meteorological observation data are also available from Taiwan CWB at https://data.gov.tw/en/datasets/9176. The meteorological observation data in Dongyan mountain were provided by Cloud and Aerosol Laboratory of the Department of Atmospheric Sciences, National Central University of Taiwan.

*Author contributions.* K.-I. L. wrote the first manuscript, performed the formal analysis, software coding, and visualization for this research article. K.-S.C. and S.-H. W. provided resources, methods, supervised, and edited the paper. K.-I. L., K.-S. C., S.-H. W., L.-H. C., Y.-C. L., P.-L. L., W.-Y. C., H.-J. C., and Y.-H. C. interpreted and discussed the data results. All authors contributed to the final paper.

*Competing interests.* The authors declare no conflicts of interest.

*Acknowledgements.* This research was supported by National Science and Technology Council 111-2625-M-008-014 and National Chung-Shan Institute of Science and Technology.

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

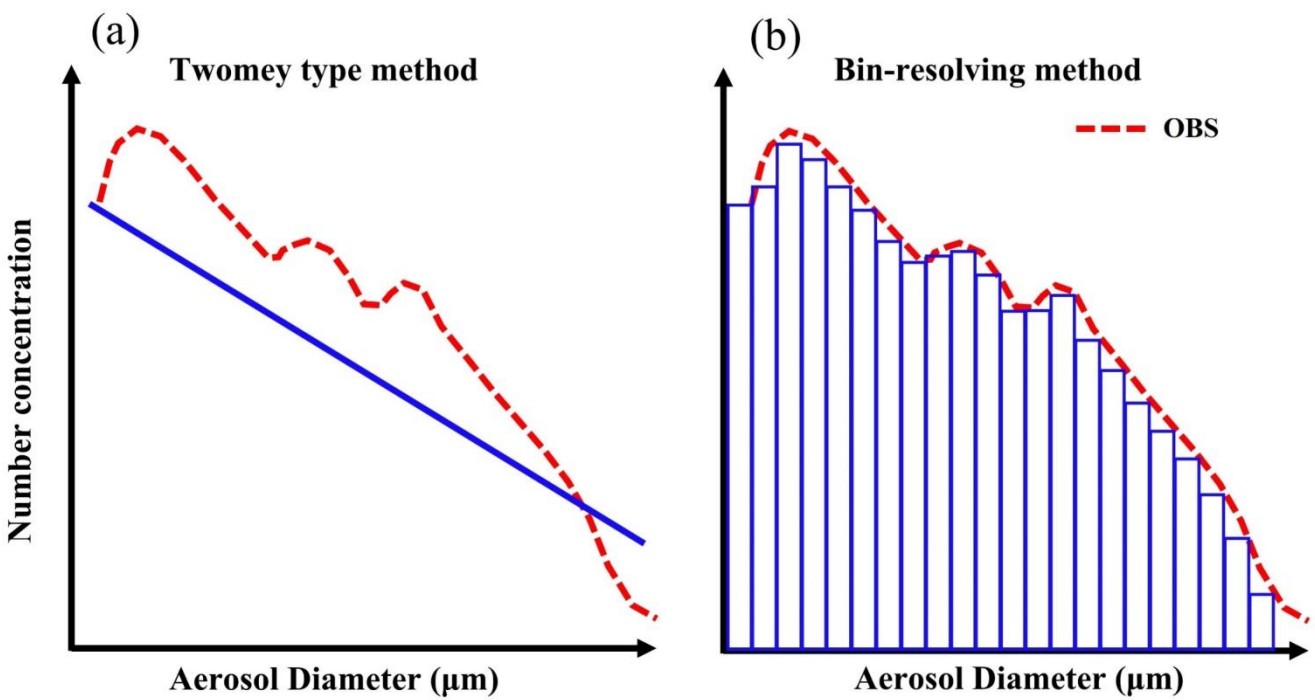

**Figure 1: The schematic illustrates two methods for determining aerosol size distribution: (a) the Twomey method is used in**
**WDM6, and (b) the bin-resolving method is used in WDM6–NCU. The blue line and bars present two different methods that**
        **model simulation describes the CCN size distribution, and the red dashed line shows as the observational CCN size**
        **distribution.**





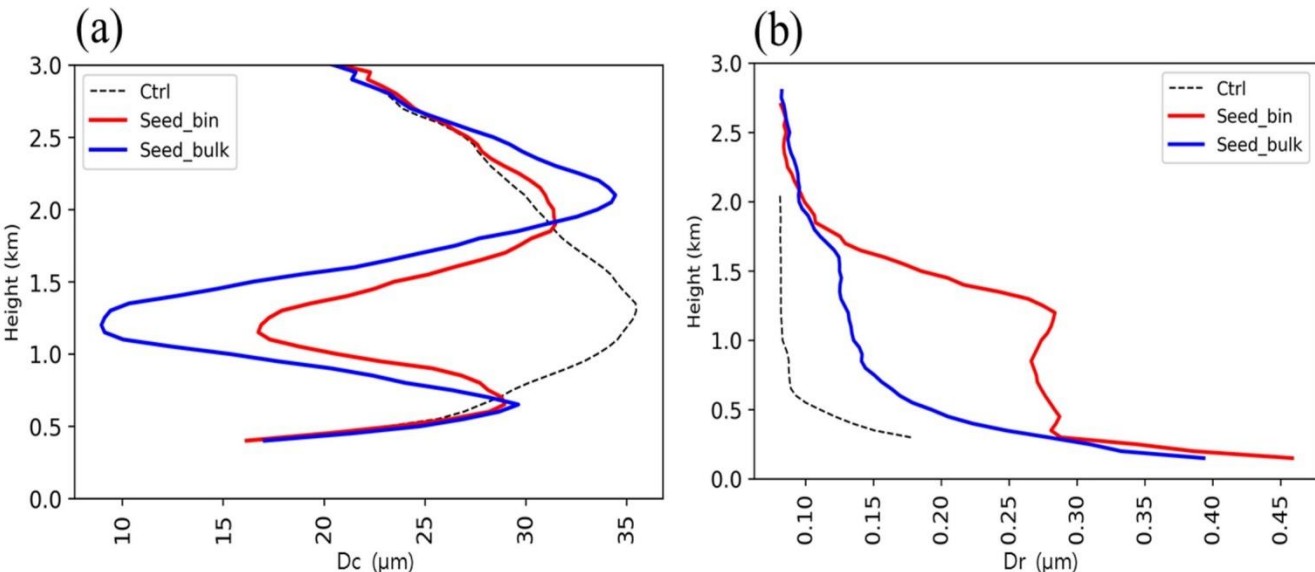

**Figure 2: Vertical profile of average (a) Dc and (b) Dr of the control run (Ctrl: without cloud seeding) and two seed runs seeding by bin and bulk methods with same number concentration (200 # cm$^{-3}$). The GCCN size distribution of Chemical Systems Research Division (CSRD: shown in section 3.2) was seeded by bin method, and fixed size CCNs with 1.2 μm (the mean diameter of GCCNs in CSRD) were seeded by bulk method.**

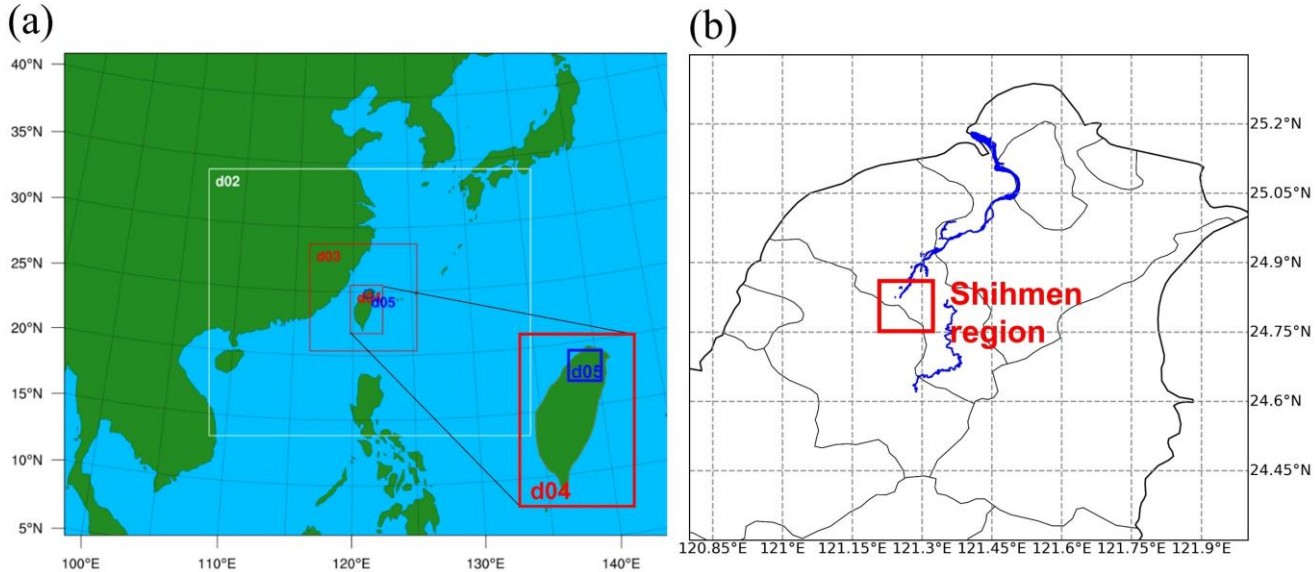

**Figure 3: (a) setting of the nested domain and construction of five nested domains. (b) location of the Shihmen region. The red rectangle in (b) represents the Shihmen region, and the rivers shown on the map are Dahan river and Danshui river.**



**Figure 4: CCN size distribution (red line) based on the observation result (black line), which is employed in the model simulation which follows the lognormal distribution with the constraining parameters of each mode for the total number concentration (N), geometric mean diameter (D), and geometric standard deviation (σ) listed in the text box**




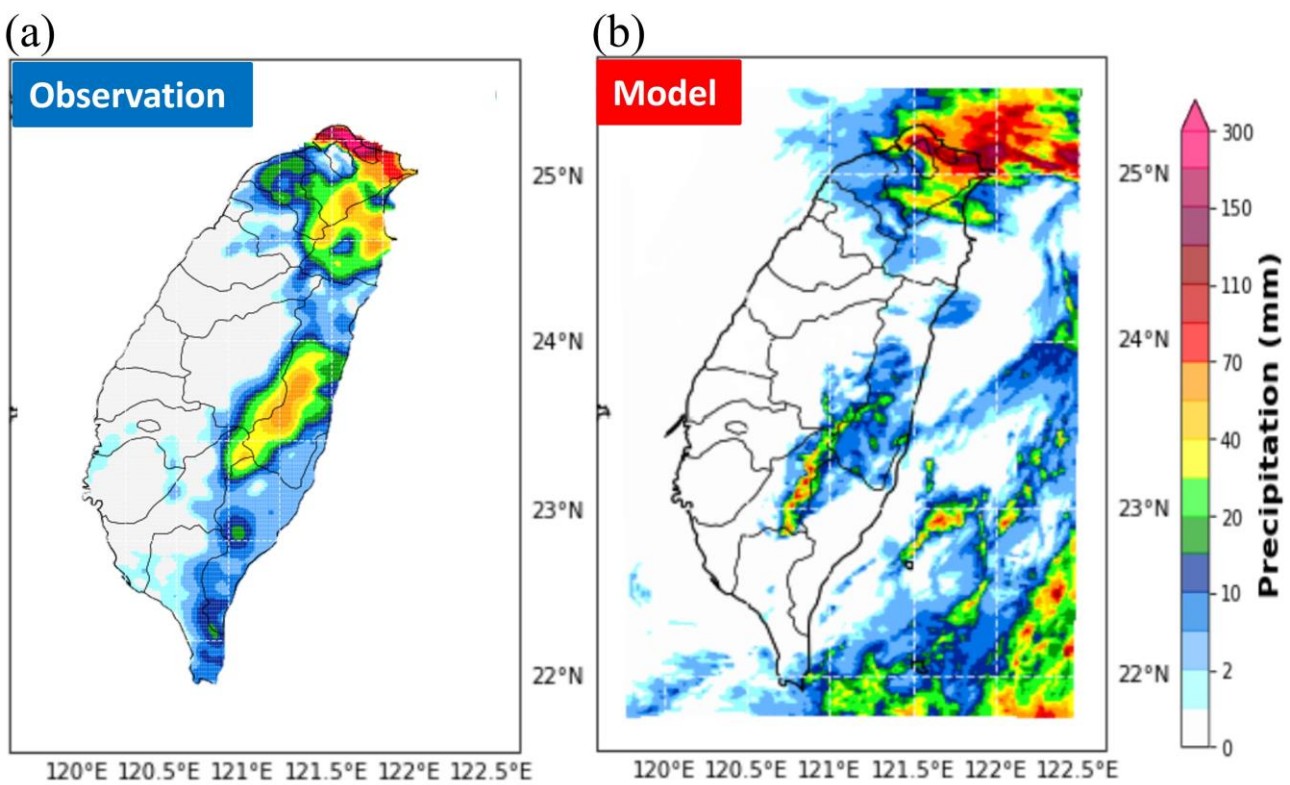

**Figure 5: Cumulative rainfall based on (a) observation data and (b) model simulation from 21 October, 2020 12:00 UTC to 22 October, 2020 12:00 UTC.**




**Figure 6: RCWF and RCSL radar reflectivity at different altitudes (1.5, 2, 3, and 5 km) and different times (06:30, 07:00, 07:30, and 08:00 UTC).**





**Figure 7: Time series (LST) of temperature, pressure, and water vapour mixing ratio based on (a) observation and (b) model simulation in the Dongyan mountain site before cloud seeding was conducted.**



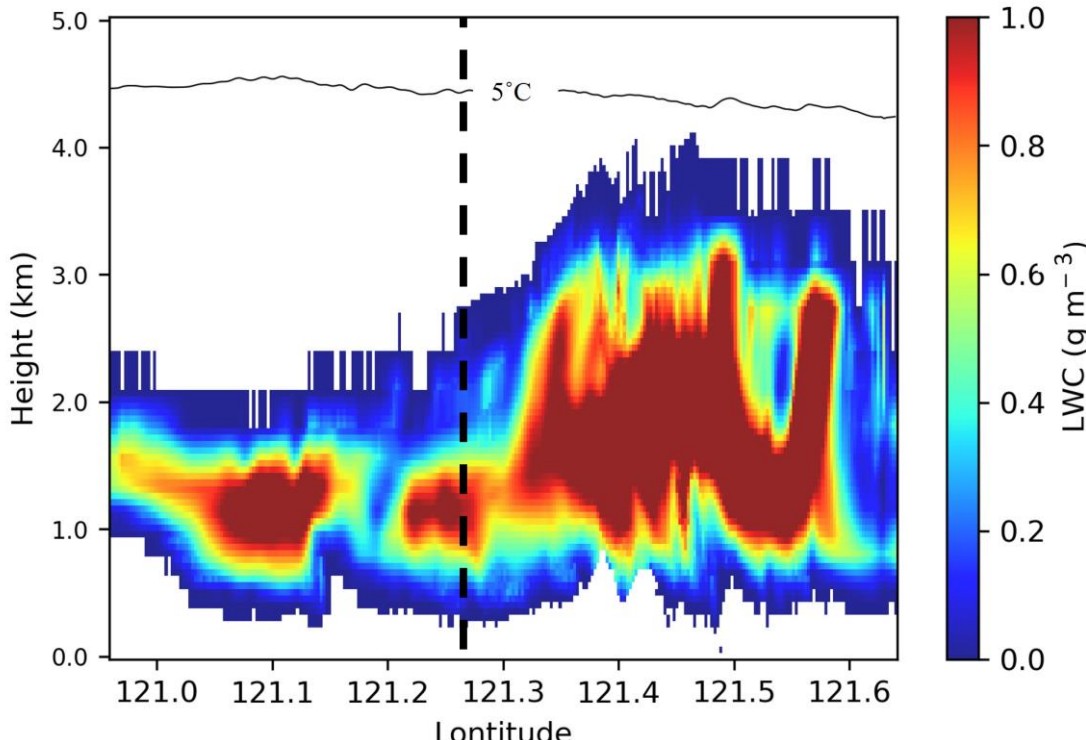

**Figure 8: Meridional mean (0.1˚ of latitude crosses the Shihmen region) of liquid water content (LWC) at 06:30 UTC on 22 October 2020. The black dashed line indicates the longitude of the Shihmen region, and the black line presents the altitude of 5 ˚C.**






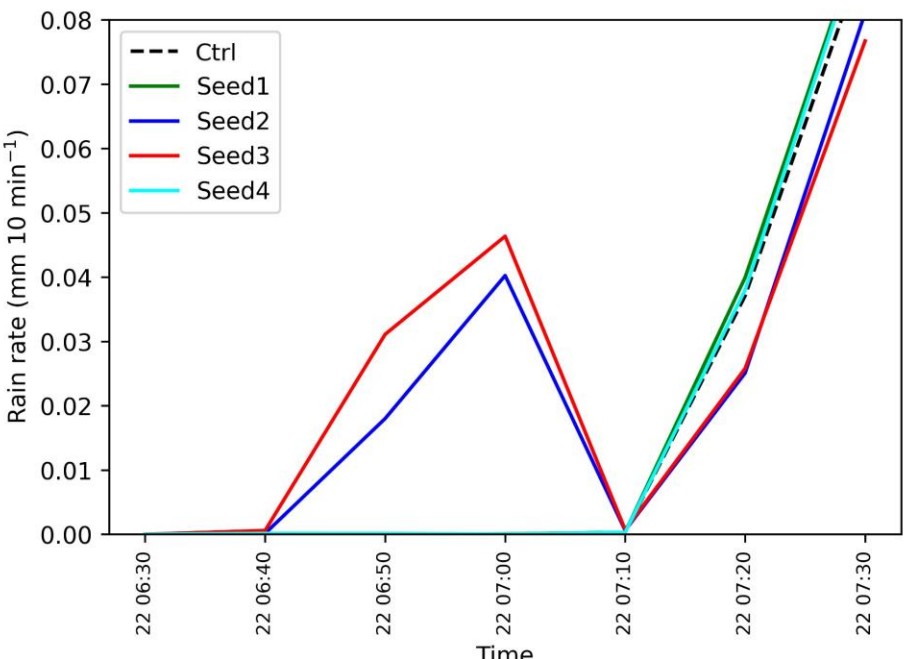

**Figure 9: Time series (UTC) of averaged rain rate of the rainy grids in the Shihmen region for seed runs and the control run.**

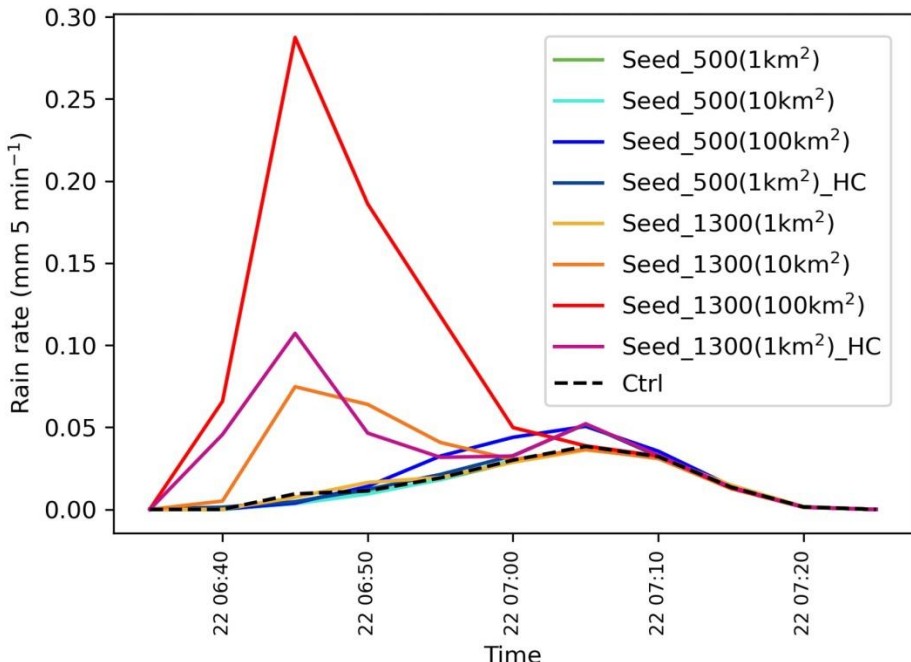

**Figure 10: Time series (UTC) of averaged rain rate of the rainy grids in the Shihmen region for seed runs and the control run.**






**Figure 11: Vertical profile of the averaged difference between the control run and seed runs of (a)–(c) log10 of CCN concentration, (d)–(f) mixing ratio of cloud (QCLOUD), (g)–(i) mixing ratio of rain (QRAIN), and (j)–(l) mean–volume–drop diameter of precipitation (Dr) after cloud seeding (started at 06:30 UTC). Warm colours represent seeding at 1300 m and cold colours represent seeding at 500 m.**



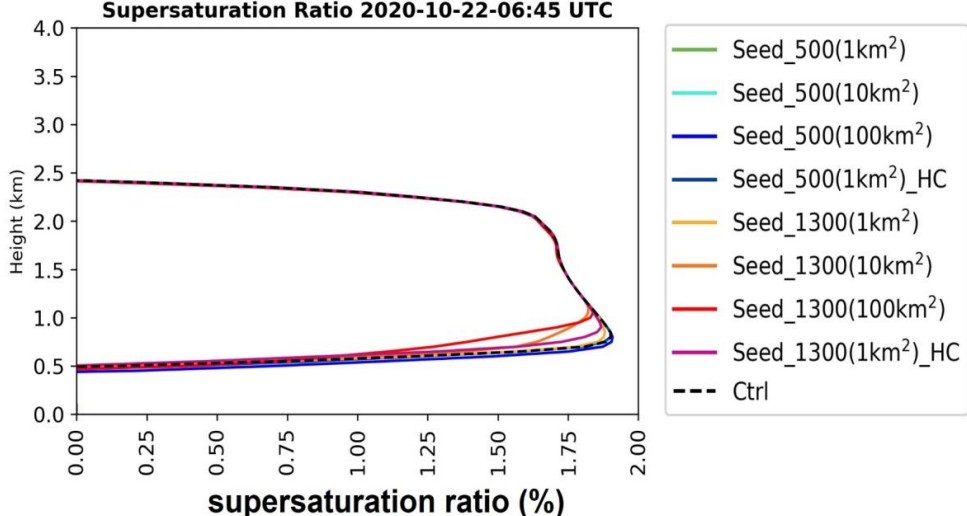


**Figure 12: Vertical profile of averaged supersaturation ratio in the Shihmen region 15 min after cloud seeding.**

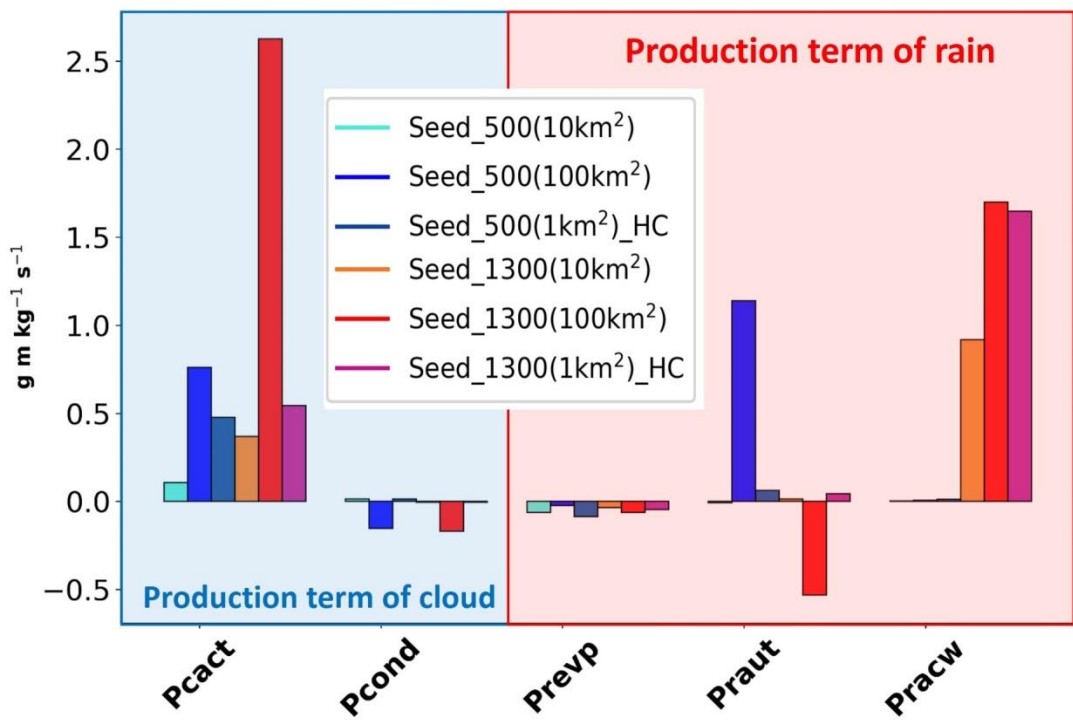

**Figure 13: Integration of simulation with height below 5 km for the averaged difference between the control run and seed runs of each parameter ($P_{cact}$, $P_{cond}$, $P_{revp}$, $P_{raut}$, and $P_{racw}$). Blue and red shaded areas represent the production term of cloud and rain, respectively**




**Figure 14: Vertical profile of the averaged difference between the control run and seed runs of the (a)–(c) cloud activation process (Pcact), (d)–(f) autoconversion process of rain (Praut), and (g)–(i) accretion process of rain (Pracw) after cloud seeding (started at 06:30 UTC). Warm colours represent seeding at 1300 m, and cold colours represent seeding at 500 m.**




**Figure 15: Size distribution of seeded CCNs at different times after cloud seeding. The black dashed line separates the particles larger and smaller than 0.4 μm.**





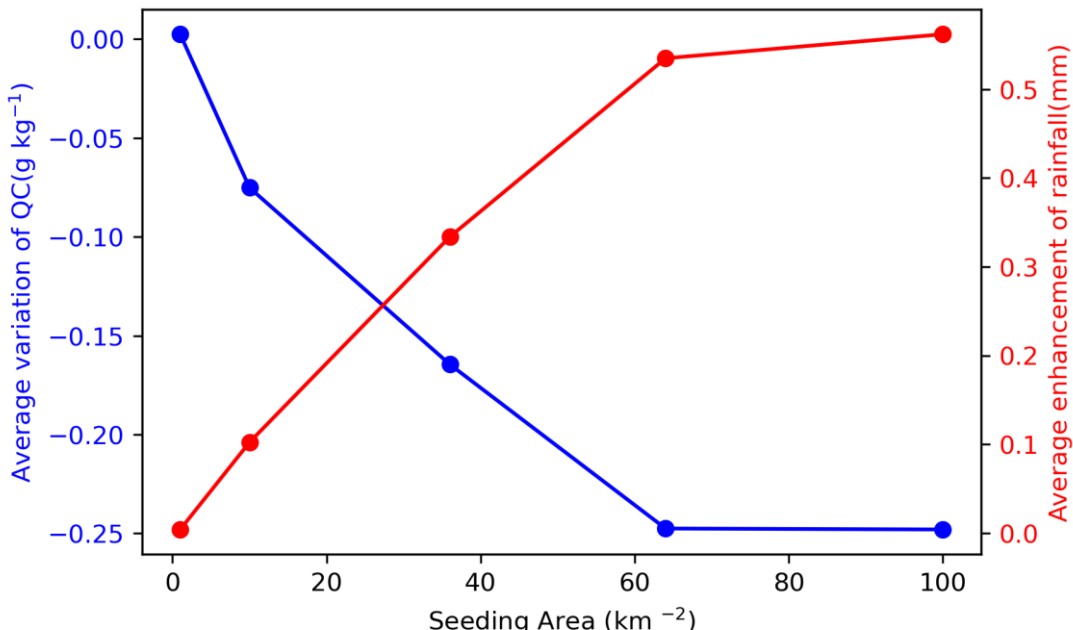

**Figure 16: Average enhancement of rain rate and average variation of cloud mixing ratio (QC) in the Shihmen region in 20 min in different seeding areas, namely 1, 10, 36, 64, and 100 km². The seeding area of 36 and 64 km² are two more runs conducted to determine the reasonable seeding area with the most effective increment of precipitation.**

**Table 1: Model configuration.**

| **WRF 3.9.1** | **D01** | **D02** | **D03** | **D04** | **D05** |
|---|---|---|---|---|---|
| Horizontal resolution | 27 km | 9 km | 3 km | 1 km | 333 m |
| Timestep | 90 s | 30 s | 10 s | 10/3 s | 1 s |
| Vertical level | 52 eta levels | | | | |
| Microphysics scheme | WDM6_NCU | | | | |
| PBL scheme | YSU | | | | LES |
| Initial and boundary condition | NCEP FNL (0.25˚× 0.25˚) | | | | |



**Table 2: Experimental design in domain four (D04).**

| Experiment | Description | Seeding area (km$^2$) | Seeding height (η/m) |
|---|---|---|---|
| Ctrl | Normal aerosol concentration | none | None |
| Seed 1 | Seed CCN (with CSRD size distribution) into a certain region. | 1 km$^2$ | 0.9865/500 m |
| Seed 2 | | | 0.9365/1000 m |
| Seed 3 | | | 0.905/1300 m |
| Seed 4 | | | 0.824/2200 m |

**Table 3: Experimental design in domain five (D05).**

| Experiment | Description | MUL factor of Concentration | Seeding area (km$^2$) | Seeding height (η/m) |
|---|---|---|---|---|
| Ctrl | Normal aerosol concentration | none | none | None |
| Seed_500(1km$^2$) | Seed CCN (with CSRD size distribution) into a certain region. | ×1 | 1 km$^2$ | 0.9865/500 m |
| Seed_500(10km$^2$) | | | 10 km$^2$ | |
| Seed_500(100km$^2$) | | | 100 km$^2$ | |
| Seed_500(1km$^2$)_HC | | ×100 | 1 km$^2$ | |
| Seed_1300(1km$^2$) | | ×1 | 1 km$^2$ | 0.905/1300 m |
| Seed_1300(10km$^2$) | | | 10 km$^2$ | |
| Seed_1300(100km$^2$) | | | 100 km$^2$ | |
| Seed_1300(1km$^2$)_HC | | ×100 | 1 km$^2$ | |