# Peer review of "Evaluation of hygroscopic cloud seeding in warm rain process by a hybrid microphysics scheme on WRF model: a real case study"

_EGUsphere, 2023_

## Author Response (AR1)

**We appreciate the suggestion from the referee. Reviewer reports are marked in black font, our responses are marked in black bold font, and the changes to the revised manuscript are marked in blue bold font.**

This research incorporates a bin method into a bulk microphysical scheme (WDM) to calculate the warm rain process and uses this scheme to investigate the warm rain seeding effect through several sensitivity experiments. I suggest that the authors should provide more information about this new scheme.

**We greatly appreciate the care taken by the reviewer in evaluating the manuscript. We believe the actions we have taken to address the comments have substantially strengthened the revision of the manuscript. Our bolded responses appear below the reviewer comments.**

1. Line 65: "The WRF model employs … the third-order Runger-Kutta numerical method for solving the time split integration of the governing equation". In fact, the third Ruger-Kutta method is just an option in the namelist. input, you can also choose the fifth-order one, so it is not "The WRF model employs", but you choose this third-order method. Please reword.

   **Response: We followed reviewer's suggestions to revise the manuscript. The sentence on Line 65 changes to**

   **"The WRF model employs an eta coordinate, which allows the grids to follow the complex terrain, and it is applied the third-order Runge–Kutta numerical method for solving the time split integration of the governing equation".**

2. Line 98: "the bins of CCNs whose size extends the critical radius will be able to activate the corresponding liquid bins", How to understand "corresponding liquid bins"? I feel that the WDM-NCU scheme has not been introduced in detail in this manuscript. What is the size range of the 43 bins? Which microphysics processes are calculated in this bin scheme? Can this scheme reflect the fact that large CCN becomes large cloud droplets? How does the scheme deal with the coupling of the bin part and bulk part?

   **Response: In WDM6-NCU, there are 43 mass-doubling aerosol bins, the radius ranging from 0.001 to 20 μm, and 43 mass-doubling liquid bins, the radius ranging from 2 to 32700 μm. After evaluating the critical radius based on Köhler theory, CCNs with the size extending critical radius will be activated to the five times CCN radius liquid bins (Lee and Baik, 2018). Activation process would be parameterized by evaluating the number concentration and mixing ratio of CCN extending the critical radius. Therefore, this scheme can reflect the fact that large CCN becomes large cloud droplets, and the activation process can contribute to the number concentration and mixing ratio of cloud and rain to couple the bin part to the bulk part. We add the detail information and the schematic of WDM-NCU in the revised manuscript (Lines 92, 99, 105-107, Fig. 2).**

**Line 92: "… the seeded CCNs are described using mass-doubling aerosol bins of 43 sizes, the radius ranging from 0.001 to 20 μm, to evaluate the effects of CCN."**

**Line 99: "… the bins of CCNs whose size extends the critical radius will be able to activate the corresponding five times CCN radius liquid bins, the radius ranging from 2 to 32700 μm (Lee and Baik, 2018)."**

**Line 105-107: "The schematic of WDM6–NCU is shown in Fig. 2. Therefore, this scheme can reveal the fact that large CCN becomes large cloud droplets, and the activation process can contribute to the number concentration and mixing ratio of cloud and rain to couple the bin part to the bulk part."**

[Figure]

**Figure 2: The schematic of WDM6-NCU.**

3. Line 102: "After the number concentration and mixing ratio of the liquid bins is calculated, they are used in the calculation of the mixing ratio and number concentration of cloud and rain, and the microphysics processes continue as the case in the original WDM6". Do you mean that the bin part of the WDM-NCU scheme only calculates the nucleation process? How do you separate the cloud and rain categories from the liquid bins? The authors have to provide more information.

**Response: WDM6-NCU aims to improve the activation process to evaluate the cloud-seeding effects more accurately. Thus, the bin part evaluates the activation process and couple with the other microphysics processes in bulk part. We use the criterion of 40 μm to separate clouds and rain droplets in the liquid bins, and this information is added in the revised manuscript (Line 104).**

**Line 104: "After the number concentration and mixing ratio of the liquid bins are calculated, they are used in the calculation of the mixing ratio and number concentration of cloud (radius ≤ 40 μm) and rain (radius > 40μm), …"**

4. Fig. 5 The simulated accumulated precipitation comes from which domain? Is the horizontal space resolution close to the observed data?

**Response: Fig. 5 results come from domain four. The rain gauge data was interpolated to the same resolution as the model simulation. We add the information of accumulative data in the illustration of Fig. 6 in the revised manuscript.**

**The illustration of Fig. 6: "Figure 6: Cumulative rainfall based on (a) rain gauge observation data and (b) model simulation (D04) from 21 October, 2020 12:00 UTC to 22 October, 2020 12:00 UTC. Observation data is interpolated to the same resolution as the model simulation."**

5. Fig. 6 What is RCWF and RCSL? They have not been explained in the manuscript. And which are observed data and which are simulated results?

   **Response: RCWF is the S-band radar at WuFen Mountain, and RCSL is the C-band radar at ShuLin. Fig. 6 shows the observed data, proving the rainfall system mainly involves warm rain processes. The model simulation also captures this feature in Fig. 8. We add the information of two radars in the illustration of Fig. 7 in the revised manuscript.**

   **The illustration of Fig. 7: "Figure 7: RCWF, the S-band radar at WuFen Mountain, and RCSL, the C-band radar at ShuLin, radar reflectivity at different altitudes (1.5, 2, 3, and 5 km) and different times (06:30, 07:00, 07:30, and 08:00 UTC)."**

6. Fig. 7, I suggest to depict the simulated results using dashed lines and put them over the observed data. So that only three 3 subplots are needed. And the Dongyan Mountain site should be plotted in Fig. 3 (or 5, 6).

   **Response: We followed reviewer's suggestions to revise the manuscript. We plot simulation data as dashed lines and observation data as solid lines in the revised manuscript (Fig. 8). The location of Dongyan mountain site is also plotted in the revised manuscript (Fig. 4).**

[Figure]

**Figure 8: Time series (LST) of temperature, pressure, and water vapour mixing ratio based on observation (solid lines) and model simulation (dashed lines) in the Dongyan mountain site before cloud seeding was conducted.**

[Figure]

**Figure 4: (a) setting of the nested domain and construction of five nested domains. (b) location of the Shihmen region. The red rectangle in (b) represents the Shihmen region, and the rivers shown on the map are Dahan river and Danshui river. The black marker X presents the location of the Dongyan Mountain site.**

7. Fig. 8: Observation or simulation?

   **Response: Fig. 8 is the simulation result. We add the description in Line 190 and the illustration of Fig. 9 in the revised manuscript.**

Line 190: "Figure 9 presents the meridional mean (0.1° latitude across Shihmen) of liquid water content (LWC) of model simulation at 06:30 UTC on 22 October, 2020."

The illustration of Fig. 9: Figure 9: Meridional mean (0.1° of latitude crosses the Shihmen region) of liquid water content (LWC) of model simulation at 06:30 UTC on 22 October 2020. The black dashed line indicates the longitude of the Shihmen region, and the black line presents the altitude of 5 °C.

8. Where are the seeding points or areas when you seed in 1km², 10km², and 100km²? These seeding points should be plotted. The seeding effects should also be shown in horizontal shadow figures, not just be plotted in lines as in Figs 9 and 10. And how much sea salt is seeded?

   **Response: We followed reviewer's suggestions to revise the manuscript. The seeding areas are centered on 24.81°N 121.26°E, plotted as a red dot in Fig. 4 of the revised manuscript. The seeding effect on surface rainfall is plotted as shadow figures in Fig. 10a and 11a of the revised manuscript. The normal seeding rate is $2.03\times10^4$ (# cm$^{-3}$ s$^{-1}$) based on the observation. The seeding rate is added in the revised manuscript (Line 155).**

[Figure]

**Figure 4: (a) setting of the nested domain and construction of five nested domains. (b) location of the Shihmen region. The red rectangle in (b) represents the Shihmen region, and the rivers shown on the map are Dahan river and Danshui river. The black marker X presents the location of the Dongyan Mountain site.**

[Figure]

**Figure 10: (a) 1 hr rain rate variation after cloud seeding (b) Time series (UTC) of averaged rain rate of the rainy grids in the Shihmen region for seed runs and the control run. The red rectangle in (a) represents the Shihmen region, and the rivers shown on the map are Dahan river and Danshui river.**

[Figure]

**Figure 11: (a) 1 hr rain rate variation after cloud seeding (b) Time series (UTC) of averaged rain rate of the rainy grids in the Shihmen region for seed runs and the control run. The red rectangle in (a) represents the Shihmen region, and the rivers shown on the map are Dahan river and Danshui river.**

**Line 155: "…, hygroscopic cloud seeding is conducted using drones that carry 10 CSRD flare seeding agents in 10 min (seeding rate: $2.03\times10^4$ # cm$^{-3}$ s$^{-1}$), …"**

9. Line 240: Why do you say seeding in such large area (100 km²) at 500 m is "impractical and ineffective" but at the same time seeding in 100 km2 at 1300 m is "more suitable"? I think both of them are impractical.

**Response: Line 240 mentioned seeding at 1300 m (in-cloud area) is more suitable because it can more effectively enhance precipitation even in smaller seeding areas, not only 100 km² (10 km×10 km). We agree that seeding in a 10 km×10 km area may be too large to execute cloud seeding. Thus, we discuss the most effective seeding area in session five of the manuscript.**

10. Fig. 13, Why Praut increases in seed_500(100 km2) but Qr does not increase as shown in Fig.11 g, h.

**Response: In fact, Qr of seed_500(100 km$^2$) also increases in Fig. 11 g and h (Fig. A1 a, b), but the value is much smaller than seed_1300(100 km$^2$). In addition, we find that the evaporation of rain is stronger at 1 to 3 km in seed_500(100 km$^2$) (Fig. A1 c, d), which might be one of the reasons there is a little Qr enhancement.**

[Figure]

**Figure A1: Vertical profile of the averaged difference between the control run and seed runs of the (a)–(b) mixing ratio of rain (QRAIN), (c)–(d) evaporation of rain (Prevp). Blue line represents Seed_500(100km$^2$), and red lines represent Seed_1300(100km$^2$).**

11. Fig. 15, Why the units of dN/dlog(D) is % ?

**Response: The number concentration of each bin is divided by the total number concentration for the percentage of each aerosol bin. The y-label of Fig. 16 in the revised manuscript is revised to "percent (%)".**

[Figure]

**Figure 16: Size distribution of seeded CCNs at different times after cloud seeding. The black dashed line separates the particles larger and smaller than 0.4 μm.**

12. Line 261: Why do you think when the seeding area is larger than 64km2, the Shihmen region no longer has plenty of cloud water to transform to precipitation? Is there any evidence supporting your conclusion? Didn't you say that seeding leads to an increase in Qc in 10 min?

    **Response: This study focuses on the Shimen region defined as a 10 km×10 km area surrounding the Shimen reservoir. In Fig. 16, the variance of Qc is similar in seeding at 64 and 100 km², which presents almost the same amount of Qc that can be transformed to Qr in the Shihmen region. Indeed, seeding can increase Qc, however, for Shihmen region, seeding in 64 and 100 km² actually have similar Qc enhancement in 10 min after cloud seeding, and consume almost the same amount of Qc to produce precipitation (Fig. A2).**

[Figure]

**Figure A2: Vertical profile of the averaged difference between the control run and seed runs of the (a)–(c) mixing ratio of cloud (QCLOUD), (d)–(F) mixing ratio of rain (QRAIN). Black line represents Seed_1300(64km2), and red lines represent Seed_1300(100km2).**

**Reference:**

Lee, H., and Baik, J.-J.: A Comparative Study of Bin and Bulk Cloud Microphysics Schemes in Simulating a Heavy Precipitation Case, Atmosphere, 9, 10.3390/atmos9120475, 2018.

Anonymous Referee #2

**We appreciate the suggestion from the referee. Reviewer reports are marked in black font, our responses are marked in black bold font, and the changes to the revised manuscript are marked in blue bold font.**

The manuscript "Evaluation of hygroscopic cloud seeding in warm rain process by a hybrid microphysics scheme on WRF model: a real case study" develops a hybrid cloud-seeding microphysics scheme and selects a case with optimal model performance and a typical weather condition in northern Taiwan to conduct a series of cloud-seeding sensitivity tests. The results show that seeding at in-cloud regions with an appropriate CCN size distribution can yield greater rainfall and that spreading the seeding agents over an area of 40–60 km$^2$ is the most efficient strategy to create a sufficient 15 precipitation rate. Overall, this is an interesting study that has the potential to advance our understanding of the evaluation of hygroscopic cloud seeding and may be of interest to ACP readers.

However, there are several issues to address before the paper is suitable to publication in ACP. Thus, I suggest minor revision before publication.

**We greatly appreciate the care taken by the reviewer in evaluating the manuscript. We believe the actions we have taken to address the comments have substantially strengthened the revision of the manuscript. Our bolded responses appear below the comments.**

Major comments:

1. It is suggested that the horizontal resolution of Large eddy simulation should be below 100m, and the horizontal and vertical resolution below the PBLH should be nearly equal. Therefore, the LES resolution of this manuscript needs to be reconsidered, unless it is well justified.

   **Response: According to Weigel et al. (2007), they applied large-eddy simulation (LES) in the horizontal grid spacing of 350 m to investigate flow features over the complex terrain. In addition, Umek et al. (2022) performed that LES with 200 m horizontal grid spacing greatly improves the stability and wind profile, compared to the non-LES mode with 1km horizontal grid spacing. Indeed, LES with finer grid resolution might produce better performance. However, our study mainly focuses on the cloud microphysics scheme and cloud microphysics properties. It is aim at making a more realistic model performance than non-LES 1 km horizontal grid**

spacing (Domain 4 in the manuscript), and we believe the LES with 333 m horizontal grid spacing (Domain 5 in the manuscript) can achieve this goal. Furthermore, the vertical grid spacing is less than 300 m below 2000 m above ground level (AGL), which is also why we applied the horizontal grid spacing of 333 m on LES.

2. As mentioned in the manuscript, the environment is crucial for cloud seeding. However, this manuscript only discusses saturation and does not delve into the discussion of the environment. Meanwhile, it only simulates a real case, so whether this conclusion has universality remains to be discussed.

**Response: In the manuscript, we focus on discussing supersaturation ratio and liquid water content (LWC) status before executing cloud seeding, because both of them are the direct factors to affect cloud microphysics. In our research, we aim at executing cloud seeding in the stratiform systems caused by the prevailing wind during fall and winter in northern Taiwan, this type of system is relative consistent and steady. Therefore, we believe the conclusion will be universal in the stratiform systems in northern Taiwan. We also have simulated a cold-front event on 29 April, 2021, and execute cloud seeding in which the environment accords with the criteria in the manuscript (the supersaturation ratio is more than 0.5% and LWC is higher than $0.6 \text{ g m}^{-3}$, and seeding area of $60 \text{ km}^2$). The result also shows the obvious enhancement of precipitation (Fig. A1). Therefore, based on this research and above response, we show that the modified microphysics scheme (WDM6-NCU) can successfully simulate the warm-cloud seeding effects, and the criteria of cloud seeding also works in another event.**

**According to Chen et al. (2020), turbulence also plays an important role to enhance the collision growth of droplets. However, in our research, we aim at executing cloud seeding in the stratiform systems in northern Taiwan, and these systems are relatively steady and do not have strong updrafts. Thus, in our research, we mainly focus on the discussion of cloud microphysics. In the near future, more observation data will be analyzed, and the detail of the environmental condition will be discussed.**

[Figure]

**Figure A1: 1 hr rain rate variation after cloud seeding on 29 April, 2021. The red rectangle represents the Shihmen region, and the rivers shown on the map are Dahan river and Danshui river.**

Minor comments:

1. Lack of discussion and comparison with existing relevant research, some comparisons can be added in the introduction or discussion section to highlight the findings of this manuscript

   **Response: We followed the reviewer's suggestions to revise the manuscript. We add some discussion and comparison with existing relevant research in the discussion.**

   **Line 263: "The criteria of LWC is comparable to Silverman (2000), which sets that LWC should be higher than 0.5 g m$^{-3}$ to execute cloud seeding."**

   **Line 271: Furthermore, several comparable results to the previous studies are found. First of all, the precipitation signals are more intense in the runs with larger seeding domains or higher seeded CCN concentrations, and this phenomenon accords with the idealized simulation of Tonttila et al. (2021). Second, few water vapour face competition from hygroscopic particles and, therefore, the saturation state of the environment is not extremely affected by cloud seeding, this result agrees with Rosenfeld et al. (2010) and the idealized simulation of Tonttila et al. (2021). Third, the simulation of Yin (2000) also shows that more rainfall enhancement is obtained in the scenario seeding**

**above cloud base. Regarding the size of seeding agents, most of the previous studies show that hygroscopic particles larger than 1 μm (GCCN) are optimal for enhancing precipitation in warm clouds (Caro et al., 2002; Segal et al., 2004). However, this study presents that the hygroscopic particles larger than 0.4 μm might also be able to contribute to enhancing precipitation in the cloud seeding. In the future, more case studies are needed to validate this result.**

2. L30: consider whether there are recent references supporting this phenomenon

   **Response: We followed the reviewer's suggestions to revise the manuscript. We add Kueh and Lin (2013) in the reference. In addition, based on our several experiments at the Shihmen region, we also find that the altitude of the cloud base is approximately 500 m above mean sea level.**

   **Line 29: "During the dry season (i.e., October to April) in Taiwan, clouds tend to be warm and relatively thin, with a cloud base of approximately 500 m above mean sea level (Chen, 1995; Kueh and Lin, 2013)."**

3. L70: Suggest not adding references in the title

   **Response: We followed the reviewer's suggestions to revise the manuscript. We remove the reference in the title.**

4. It is recommended to merge Figures 7a and b for comparison purposes

   **Response: We followed the reviewer's suggestions to revise the manuscript. We plot simulation data as dashed lines and observation data as solid lines in the revised manuscript (Fig. 8).**

[Figure]

**Figure 8: Time series (LST) of temperature, pressure, and water vapour mixing ratio based on observation (solid lines) and model simulation (dashed lines) in the Dongyan mountain site before cloud seeding was conducted.**

5. Figure 7: The horizontal axis does not appear to be a font

   **Response: We followed the reviewer's suggestions to revise Fig. 7 to Fig. 8, shown above, in the revised manuscript.**

6. Figure 8: "Lontitude" "Longitude"

   **Response: We followed the reviewer's suggestions to revise Fig. 8 to Fig. 9 in the revised manuscript.**

[Figure]

**Figure 9: Meridional mean (0.1° of latitude crosses the Shihmen region) of liquid water content (LWC) of model simulation at 06:30 UTC on 22 October 2020. The black dashed line indicates the longitude of the Shihmen region, and the black line presents the altitude of 5 °C.**

7. Figure 14: maintaining a unified time zone for the entire manuscript

**Response: We followed the reviewer's suggestions to revise the manuscript. Only UTC is used in the revised manuscript.**

**Reference:**

Chen, S., Xue, L., and Yau, M.-K.: Impact of aerosols and turbulence on cloud droplet growth: an in-cloud seeding case study using a parcel–DNS (direct numerical simulation) approach, Atmospheric Chemistry and Physics, 20, 10111-10124, 10.5194/acp-20-10111-2020, 2020.

Kueh, M.-T. and Lin, P.-L.: Springtime cloud properties in the Taiwan Strait: synoptic controls and local processes, Theoretical and Applied Climatology, 116, 463-480, 10.1007/s00704-013-0969-y, 2013.

Umek, L., Gohm, A., Haid, M., Ward, H. C., and Rotach, M. W.: Influence of grid resolution of large-eddy simulations on foehn-cold pool interaction, Q J R Meteorol Soc, 148, 1840-1863, 10.1002/qj.4281, 2022.

Weigel, A. P., Chow, F. K., and Rotach, M. W.: On the nature of turbulent kinetic energy in a steep and narrow Alpine valley, Boundary-layer meteorology, 123, 177-199, 2007.

---

## Referee Report (RR1)

The authors carefully respond to my comments and revise the manuscript. In the revised manuscript, the WDM-NCU scheme is introduced in more detail. We appreciate the efforts made by the authors, although we still have some new questions:

1. If I understand correctly, when nucleating, a CCN particle will become a liquid droplet whose radius is five times larger than the CCN particle. As the radius of CCN particles ranges from 0.001 to 20 um, it means a CCN particle with the radius of 20 um will become a liquid particle with the radius of 100 um. That is to say, a large CCN particle will directly become a rain drop immediately after nucleation! It is impossible! There must be an upper limit on the radius that the newly nucleated particles will be.

2. Based on your description in the revised manuscript, it seems that only the nucleation process is calculated using the BIN part of the scheme, the mass and number concentration of the newly nucleated particles are added to the BULK part of this scheme and all the other warm rain processes are calculated in the BULK part. In the BIN part, it is true that "large CCN becomes larger liquid particle". But as you only add the mass and number concentration of the newly nucleated particles to the BULK part, it ("large CCN becomes larger liquid particle") may not be true, since the particle size distribution (PSD) of liquid water in the bulk part may deviate from the that in the bin part. You should compare the

PSDs in these two parts after nucleation to see whether the parameterized PSDs in the BULK part are close to that in the BIN part.

3. Line 21. "Taiwanese government" is not a globally accepted term, and I don't see the necessity of using "Taiwanese government" here, please reword.

---

## Author Response (AR2)

**We appreciate the suggestion from the referee. Reviewer reports are marked in black font, our responses are marked in black bold font, and the changes to the revised manuscript are marked in blue bold font.**

The authors carefully respond to my comments and revise the manuscript. In the revised manuscript, the WDM-NCU scheme is introduced in more detail. We appreciate the efforts made by the authors, although we still have some new questions:

**We greatly appreciate the care taken by the reviewer in evaluating the manuscript. We believe the actions we have taken to address the comments have substantially strengthened the revision of the manuscript. Our bolded responses appear below the reviewer comments.**

1. If I understand correctly, when nucleating, a CCN particle will become a liquid droplet whose radius is five times larger than the CCN particle. As the radius of CCN particles ranges from 0.001 to 20 um, it means a CCN particle with the radius of 20 um will become a liquid particle with the radius of 100 um. That is to say, a large CCN particle will directly become a rain drop immediately after nucleation! It is impossible! There must be an upper limit on the radius that the newly nucleated particles will be.

   **Response: According to Lee and Baik (2018) and Kogan (1991), the maximum radius of aerosols is 2 μm and 7.6 μm, and after activation, the activated CCNs will turn into about five times radius droplets. In WDM6-NCU, to consider the effects of more giant CCNs, the maximum radius of aerosols is set as 20 μm. Regarding the effects of giant CCNs (GCCNs), Posselt and Lohmann (2008) also depicted the process that GCCNs are directly activated to raindrops. On the other hand, we agree with the opinion of the referee that there must be an upper limit or the smaller growth rate for the CCNs with the radius larger than 7.6 μm. In this research, there are almost without CCNs larger than 2 μm (Fig. 5 in the manuscript). In addition, we check the size distribution of the liquid bin (Fig. 1A: the unit is percentage) which shows that nearly no CCNs are directly activated to raindrops. In the future, an upper limit or the smaller growth rate for the CCNs with the radius larger than 7.6 μm will be added in WDM6-NCU. We greatly appreciate the suggestion of the referee.**

[Figure]

   **Figure 1A: the size distribution of liquid bin 10 min after cloud seeding. The black dashed line presents the diameter that separates cloud and raindrops.**

2. Based on your description in the revised manuscript, it seems that only the nucleation process is calculated using the BIN part of the scheme, the mass and number concentration of the newly nucleated particles are added to the BULK part of this scheme and all the other warm rain processes are calculated in the BULK part. In the BIN part, it is true that "large CCN becomes larger liquid particle". But as you only add the mass and number concentration of the newly nucleated particles to the BULK part, it ("large CCN becomes larger liquid particle") may not be true, since the particle size distribution (PSD) of liquid water in the bulk part may deviate from the that in the bin part. You should compare the PSDs in these two parts after nucleation to see whether the parameterized PSDs in the BULK part are close to that in the BIN part.

**Response: This study attempts to extract the realistic and detailed cloud seeding information, which can be applied for the BULK part (activated CCN number concentration and mixing ratio), from the BIN part. We understand the differences between BIN and BULK parts and agree that the PSD might be different between these two parts. Figure 2A presents the PSD in the BULK part before and after the activation from the BIN part. Figure 2A shows that the BIN part PSD information will be evenly allocated to bulk parts and described by the gamma function. In the future, we should think about how to conserve more PSD information from the BIN part to the BULK part. We appreciate the suggestion of the referee and revise the manuscript:**

**Line 106: "Therefore, this scheme can reveal the fact that large CCN becomes large cloud droplets in the bin part, …"**

**We integrate the above suggestions and add some information to the discussion of the manuscript:**

**Line 281: "Regarding the WDM6-NCU scheme, there are still some areas that require improvement. First, in the bin part, to consider the effects of giant CCNs, the maximum radius of aerosols is set as 20 μm which is different from the 2 μm setting in Lee and Baik (2018) and 7.6 μm in Kogan (1991). Although, in this research, there are almost without CCNs larger than 2 μm (Fig. 5), and nearly no CCNs are directly activated to the raindrops. It might be more reasonable that an upper limit or the smaller growth rate for the CCNs with the radius larger than 7.6 μm are defined in the WDM6-NCU. Second, for the connection between the bin and bulk parts, WDM6-NCU can extract the realistic activated CCN number concentration and mixing ratio from the bin part that can be applied to the bulk part. However, the information of droplet size distribution (DSD) might be different between the bin and bulk parts. In the future, an upper limit or the smaller growth rate for the CCNs with the radius larger than 7.6 μm will be added in WDM6-NCU, and we should think about how to conserve more DSD information from the bin part to the bulk part."**

[Figure]

**Figure 2A: The PSD in the BULK part before and after the activation from the BIN part.**

3. Line 21. "Taiwanese government" is not a globally accepted term, and I don't see the necessity of using "Taiwanese government" here, please reword.

**Response: We followed reviewer's suggestions to revise the manuscript.**

**Line 21: "…, which prompted the government to identify methods to address water scarcity problems with utmost urgency."**

**Reference:**

Kogan, Y. L.: The simulation of a convective cloud in a 3-D model with explicit microphysics. Part I: Model description and sensitivity experiments, Journal of the Atmospheric Sciences, 48, 1160-1189, 1991.

Lee, H. and Baik, J.-J.: A Comparative Study of Bin and Bulk Cloud Microphysics Schemes in Simulating a Heavy Precipitation Case, Atmosphere, 9, 10.3390/atmos9120475, 2018.

Posselt, R. and Lohmann, U.: Influence of Giant CCN on warm rain processes in the ECHAM5 GCM, Atmospheric Chemistry and Physics, 8, 3769-3788, 2008.